# Anti-MOG Antibody-Associated Unilateral Cortical Encephalitis with Bilateral Meningeal Involvement: A Case Report

**DOI:** 10.3390/brainsci13020283

**Published:** 2023-02-08

**Authors:** Bo Ren, Shiying Li, Bin Liu, Jinxia Zhang, Yaqing Feng

**Affiliations:** First Department of Neurology, North China University of Science and Technology Affiliated Hospital, Tangshan 063000, China

**Keywords:** myelin oligodendrocyte glycoprotein, MOG-IgG-associated disorders, anti-MOG antibody-associated encephalitis, encephalitis, demyelinating diseases

## Abstract

A 27-year-old Han Chinese woman presented with fever, headache, lethargy, and difficulty in expression. Magnetic resonance imaging (MRI) detected extensive hyperintensity of the left-sided frontoparietal, temporal, occipital, and insular cortices via fluid-attenuated inversion recovery (FLAIR) imaging. Post-contrast MRI revealed linear enhancement in the frontoparietal, temporal, and occipital sulci bilaterally. The detection of anti-myelin oligodendrocyte glycoprotein (MOG) was positive in the cerebrospinal fluid (CSF) and serum. The patient was diagnosed with anti-MOG antibody-associated unilateral cortical encephalitis with bilateral meningeal involvement. The patient received low doses of intravenous dexamethasone followed by oral prednisone, which was tapered until withdrawal. The treatment significantly improved the patient’s symptoms. A one-month follow-up showed that the patient gradually resumed her normal lifestyle. No further relapse was recorded after a one-year follow-up. MRI performed almost a year after the initial symptom onset showed that the FLAIR signal had decreased in the left insular lobe, and the abnormal cortical signal of the FLAIR in the original left frontotemporal occipital lobe had disappeared. Thus, we report a rare case of anti-MOG antibody encephalitis (unilateral cortical encephalitis with bilateral meningeal involvement) in an adult patient. This study provides a reference for the clinical diagnosis and treatment of MOG antibody-associated unilateral cortical encephalitis.

## 1. Introduction

Myelin oligodendrocyte glycoprotein (MOG) is a myelin glycoprotein specifically expressed in the oligodendrocytes of the central nervous system. Anti-MOG antibodies activate complement cascades, leading to oligodendrocyte degradation and demyelination [1]. Positive anti-MOG antibodies characterize various inflammatory demyelinating diseases of the central nervous system. The diseases mediated by anti-MOG antibodies differ from those mediated by multiple sclerosis (MS) and water-channel aquoporin4 (AQP4) antibody-positive neuromyelitis optica spectrum disorders (NMOSD) based on pathological changes, clinical features, treatment, and prognosis. As a result, the anti-MOG antibody-mediated diseases are considered an independent class of diseases (anti-myelin oligodendrocyte glycoprotein IgG-associated disorders (MOGAD)) [2]. The clinical cases of MOG antibody-associated unilateral cortical encephalitis are rare. Herein, we report a case of an adult with anti-MOG antibody encephalitis (unilateral cortical encephalitis with bilateral meningeal involvement), to provide a reference for the clinical diagnosis and treatment of this disease.

## 2. Case Presentation

A 27-year-old Han Chinese woman was admitted to our hospital because of a headache with fever for 2 weeks and difficulty in expression for 1 day. The first symptoms included intermittent headache and fever (37.6 °C), followed by slurred speech, a clumsy tongue, and an inability to speak. The patient was slow in thinking and could only understand part of other people’s speech. The patient experienced general weakness, hypersomnia, and one episode of vomiting. There were no dysphagia, no convulsions, and no urinary or stool disorders. She had not been infected with the novel coronavirus. The patient had no personal or family history of any unique diseases. Physical examination showed that her body temperature and blood pressure were 36.5 °C and 110/62 mmHg, respectively. The patient was 163 cm in height and 55 kg in weight. Cardiopulmonary and abdominal examinations showed no abnormalities. A nervous system physical examination showed the following: lethargy, receptive and expressive aphasia, poor physical coordination, level 5 muscle strength in the limbs, normal muscle tension and tendon reflex, negative Babinski and Chaddock signs, and positive stiff-neck signs. A fundus examination showed no abnormalities. Routine blood tests including liver function, renal function, electrolyte, glucose, cholesterol, triglycerides, homocysteine, blood coagulation, thyroid hormone, rheumatoid factor, antinuclear antibodies, female tumor factor, HIV antibody, treponema pallidum antibody, hepatitis virus family, virus antibody series, and tuberculosis antibody analyses showed no abnormalities. Moreover, brain CT, chest CT, and ECG showed no abnormalities. An electroencephalogram (EEG) showed severe abnormalities (scattered spiky and spiky slow waves) (Figure 1). A brain MRI and a diffusion-weighted imaging (DWI) examination showed abnormal signals in the cortex of the left frontoparietal, temporal, occipital, and insular cortices (Figure 2A–D). A post-contrast MRI revealed linear enhancement in the frontoparietal, temporal, and occipital sulci bilaterally, indicating an inflammatory disease (Figure 2E–H). A brain magnetic resonance angiography (MRA) and venography (MRV) showed no abnormalities. A lumbar puncture analysis showed that the CSF pressure was 140 mmH_2_O, and the CFS was colorless and transparent. The total cell number, white blood cell number, and total protein in the CSF were 350 × 106/L, 10 × 106/L, and 0.46 g/L (normal range 0.08–0.32 g/L), respectively. The chloride and glucose levels were normal. The CSF cytology was abnormal and dominated by lymphocytic responses (70%) and activated monocytosis (9%). CSF ink staining, acid-fast staining, virus antibody series, tuberculosis antibody, and CSF culture were all negative. Metagenomic next-generation sequencing did not detect any pathogens in the CSF. NMDA-R-Ab, CASPR2-Ab, AMPA1-R-Ab, AMPA2-R-Ab, LGI1-Ab, GABAB-R-Ab, GAD65-Ab, and other antibodies associated with autoimmune encephalitis provided negative reactions in the blood and CSF. The AQP4 antibody and oligoclonal IgG bands (OB) also resulted in negative reactions in the blood and CSF. However, MOG antibodies reacted positively in the serum (1:10) and CSF (1:10). The patient was finally diagnosed with anti-MOG antibody-associated unilateral cortical encephalitis with bilateral meningeal involvement.

The patient then received an injection of dexamethasone sodium phosphate (15 mg) once daily for two weeks. The treatment significantly improved the patient’s speech, body movement, cognitive ability, memory, and orientation. Moreover, no positive signs were found after a neurological physical examination. After two weeks of treatment, a re-examination of the lumbar puncture showed that the CSF pressure was 150 mmH_2_O. The total white blood cells and mononuclear white blood cells were 25 × 106/L and 80%, respectively. The total protein, chloride, and glucose levels were normal. However, CSF cytology was abnormal and was dominated by a lymphocyte response (95%). Metagenomic next-generation sequencing did not detect any pathogens in the CSF. Anti-MOG antibody still yielded a positive reaction in the CSF and serum (1:10). A re-examination of the brain MRI enhancement showed that the abnormal signals in the bilateral frontoparietal, temporal, and occipital sulci were less than before (Figure 2I–L). The patient was discharged and received a gradually tapering dose of oral prednisone (60 mg/day), tapering by 5 mg every week until discontinuation. A one-month follow-up showed that the patient could live and work normally. The patient also had a normal EEG. Re-examination showed that the anti-MOG antibody provided a negative reaction in the serum. A re-examination of the routine blood tests including liver function, renal function, electrolyte, and glucose analyses showed no abnormalities. No further relapse was recorded after a one-year follow-up. The brain MRI was repeated after two months, and the results showed that the FLAIR signal decreased in the left insular lobe, and the abnormal signal was partially absorbed in the left frontoparietal, temporal, and occipital cortices (Figure 2M–P). MRI performed almost one year after the initial symptom onset showed that the FLAIR signal had decreased in the left insular lobe, and the abnormal cortical signal of the FLAIR in the original left frontal, parietal, temporal, and occipital lobes had disappeared (Figure 2Q–T).

## 3. Discussion

MOG consists of 218 amino acids and is specifically expressed in the outermost myelin sheath and oligodendrocytes of the central nervous system of mammals. It is expressed later than other myelin proteins, so it may be a marker of oligodendrocyte maturation and myelin densification. MOG can maintain the stability of the myelin structure, regulate the cytoskeleton, activate the complement, etc. [2,3]. MOG antibodies are classified as pathogenic and non-pathogenic. Pathogenic antibodies recognize a MOG epitope in the spatial stereostructure that can trigger extensive demyelination, while non-pathogenic antibodies recognize linear antigen epitopes. The current studies suggest that anti-myelin oligodendrocyte glycoprotein-IgG (MOG-IgG) may be a pathogenic antibody. The pathogenic mechanism of anti-MOG-IgG may be activated when the blood–brain barrier is broken, causing the MOG antigen to leak into the peripheral blood, activate CD4+T lymphocytes, and recruit and activate B cells to produce anti-MOG-IgG. Pro-inflammatory MOG-specific T cells and macrophages cause inflammation in the central nervous system, and MOG antibodies enter the central nervous system and bind to MOG, causing oligodendrocyte injury and demyelination [4]. Since anti-MOG-IgG is produced by B cells in the peripheral blood, serum is the preferred test sample. However, some patients indicated that the MOG antibody in the CSF reacted positively, while that in the serum reacted negatively, suggesting the existence of intrathecal synthesis of the MOG antibody. Some studies have shown that the analysis of the MOG antibody in the CSF can improve the diagnostic sensitivity of seronegative patients [5], but other studies believe that the suggested role of the MOG antibody in the CSF remains to be further evaluated [6]. In this case, the patient’s anti-MOG antibodies were present in both serum and cerebrospinal fluid, indicating damage to the blood–brain barrier. The MRI of the patient indicated pia enhancement, which confirmed the blood–brain barrier injury in the patient. The patient’s mog antibody titer was 1:10; a low titer might indicate a good prognosis and low recurrence for the patient [7].

The pathological features of MOGAD are mainly inflammatory white matter demyelinating lesions around small vessels, mostly in the cortex, accompanied by partial axon retention and reactive gliosis. CD4+T cells and granulocyte infiltration were the main lesions [8]. The quantitative balance between pathogenic T cells and MOG antibodies may be related to different phenotypes of the disease. If the antibody titer is high, it is easy to show related demyelinating lesions, otherwise, it is more likely to observe ADEM-like inflammation. Major histocompatibility complex (MHC) gene polymorphisms also affect the distribution of lesions in different locations [9]. The clinical phenotype of MOGAD can include one or several of the following: optic neuritis, myelitis, meningitis, encephalitis, etc. [10,11]. Encephalitis is a key clinical phenotype of MOGAD. MOG antibody-associated encephalitis has various clinical manifestations, including seizures, headache, disturbance of consciousness, limb weakness, diplopia, and cognitive impairment [12,13]. Budhram’s team reviewed 20 cases of MOG antibody-associated cortical encephalitis and showed that seizures (85%) are the most common symptoms, followed by headache (70%), fever (65%), and other cortical symptoms (55%) [14]. In the present case, the clinical symptoms of the patient included headache, disturbance of consciousness, aphasia, and fever, without optic nerve edema or spinal cord involvement. Although the patient did not have seizures, an EEG showed epileptiform discharges on admission. However, the EEG was normal after the clinical symptoms disappeared.

The lesions in MOG antibody-associated encephalitis are located in the cortex/subcortex [12], thalamus, hippocampus, corpus callosum, pons, cerebellum, midbrain, medulla oblongata, etc. [15]. Cortical/subcortical, thalamic, and hippocampal lesions are relatively specific. Budhram’s team reviewed 20 cases of anti-MOG cortical encephalitis and showed that most are unilateral (80%), while some involve the meninges (30%) [14]. T2-flair hyperintensity in unilateral cortex and seizures characterize MOG antibody-associated encephalitis. Herein, a brain MRI detected FLAIR hyperintensity (unilateral cortical and subcortical lesions) in the local cortex of the left insula and left fronto-parieto-temporo-occipital lobes, consistent with the MRI imaging characteristics of MOG antibody-associated encephalitis [15]. When clinical imaging suggests limbic system involvement in encephalitis, we also need to distinguish herpes simplex virus encephalitis, paraneoplastic encephalitis, Hashimoto’s encephalopathy, autoimmune encephalitis, etc. Combined with the patient’s history, early antibody detection is helpful for early diagnosis and treatment. Meningitis is also an important clinical and imaging manifestation of MOG antibody encephalitis [16]. Grace Y Gombolay’s team reported 11 patients with MOGAD with aseptic meningitis and leptomeningeal enhancement. They suggested that aseptic meningitis with leptomeningeal enhancement may be the initial symptom of MOGAD [17]. However, unilateral activating lesions with bilateral MRI meningeal contrast lesions are relatively uncommon. In this case, the brain MRI showed no enhancement of the brain parenchyma but showed a linear enhancement of the bilateral fronto-parieto-temporo-occipital sulcus. Therefore, the patient was diagnosed with anti-MOG antibody-associated unilateral cortical encephalitis with bilateral meningeal involvement. An MRI enhancement of the bilateral fronto-parieto-temporo-occipital lobe showed a line-like enhancement in the sulci, suggesting that the patient’s blood–brain barrier was broken. Since no pathogens were detected in the mNGS of this patient, the patient was presumed to have aseptic meningitis mediated by MOG antibodies. These results suggest that MOG-antibody testing should be considered in patients presenting with unexplained aseptic meningitis and unilateral cortical lesions.

To date, no large-sample research has reported on the treatment plan for MOG antibody-associated encephalitis. High-dose hormone shock therapy is recommended as the first choice for MOG antibody-associated encephalitis in the acute phase. In this study, low-dose hormone shock therapy significantly improved the clinical symptoms of the patient. Therefore, the prognosis of MOG antibody-associated corticoencephalitis may be better than that of other clinical phenotypes. In this case, although the bilateral meninges were involved, a low-dose hormone therapy still achieved a good effect, which may be related to the small brain parenchymal lesion and the relatively young age of the patient. However, long-term follow-up and more clinical data are still needed. It is the common pursuit of doctors and patients to achieve the best therapeutic effect with the smallest drug dose and the smallest number of drug side effects. A large amount of clinical data are needed to find the best treatment.

Although MOGAD typically has a monophasic course, it can also recur. Chanjira Satukijchai‘s team studied 276 patients with MOGAD and found that the 8-year risk of relapse was 36.3% [18]. The recurrent type is usually characterized by optic neuritis. However, the related risk factors for the disease recurrence are unknown [19]. Most lesions disappear during follow-up, while some have residual T2 hyperintensity [20]. Sugimoto reported a case of unilateral corticomeningoencephalitis with positive MOG antibody, which developed extensive transverse myelitis 20 days after onset [21]. In this study, recurrence did not occur after one year of follow-up. Although the FLAIR imaging showed some abnormal signals, the patient had no clinical symptoms. Of course, a longer follow-up is needed to provide a better reference for clinical treatment.

## 4. Conclusions

CSF and serum tests for MOG antibodies should be considered in patients with unexplained unilateral cortical encephalitis with headache, fever, and disturbance of consciousness, for an accurate diagnosis and timely treatment. Low-dose hormone therapy may also achieve good clinical prognosis for patients with MOG antibody-associated unilateral corticoencephalitis. We strive to achieve the best therapeutic effect with the smallest drug dose and the smallest number of drug side effects, but the long-term outcomes of low-dose hormone therapy remain unclear. The visual evoked potential (VEP) was not evoked in this patient, which was the limitation of this study. In addition, this is just a case report. Future studies with more data and longer follow-up times are needed to find the best treatment.

## Figures and Tables

**Figure 1 brainsci-13-00283-f001:**
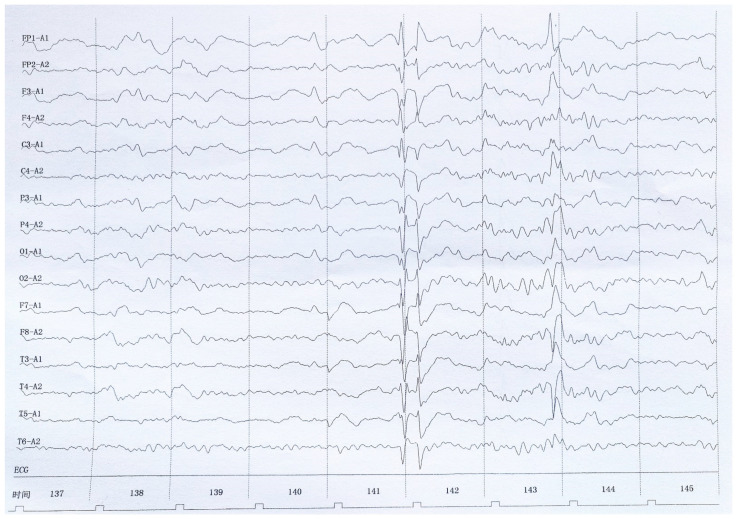
EEG showing the main rhythmic activities (low-amplitude α wave, irregular waveform, scattered sharp wave, and sharp–slow wave). The visual response did not significantly change (100 µ/cm and 30 mm/s).

**Figure 2 brainsci-13-00283-f002:**
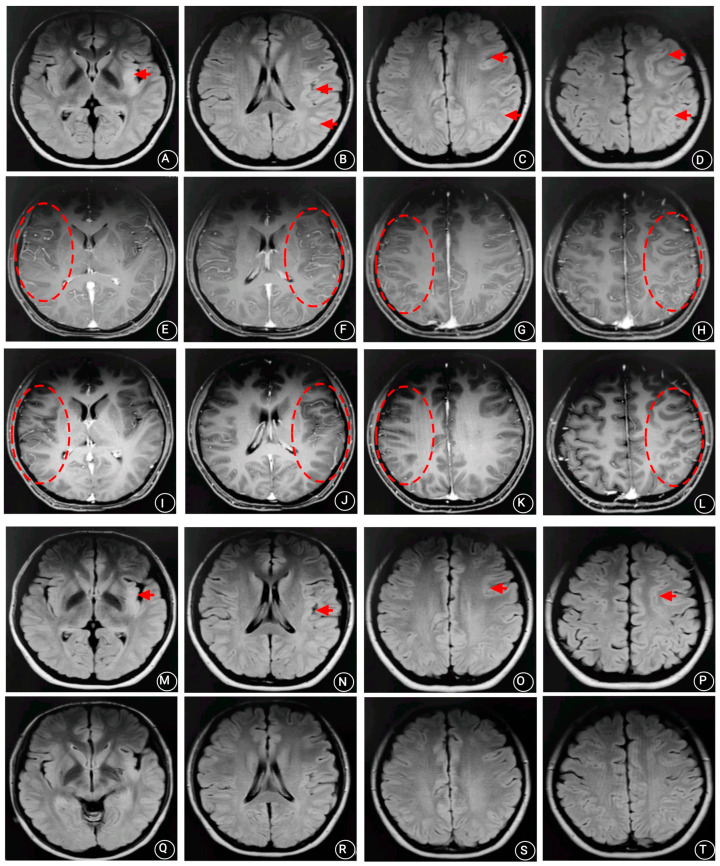
Brain MRI of the patient at admission (2 weeks after symptom onset) showing extensive hyperintensity of the left-sided frontoparietal, temporal, occipital, and insular cortices on FLAIR imaging (arrows) (**A**–**D**). Post-contrast MRI revealed linear enhancement in frontoparietal, temporal, and occipital sulci bilaterally (ellipses) (**E**–**H**). Post-contrast MRI after 2 weeks of treatment showed that the linear enhancement in these areas was less than before (ellipses) (**I**–**L**). MRI repeated after two months. The FLAIR signal had decreased in the left insular lobe, and the abnormal signal was partially absorbed in the left frontoparietal, temporal, and occipital cortices (arrows) (**M**–**P**). MRI performed almost one year after the initial symptom onset showed the FLAIR signal had decreased in the left insular lobe, and the abnormal cortical signal of FLAIR has disappeared in the original left frontal, temporal, parietal, and occipital lobes (**Q**–**T**).

## Data Availability

The data generated for this study are available on request to the corresponding author.

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
