# Peer review of "Anti-MOG Antibody-Associated Unilateral Cortical Encephalitis with Bilateral Meningeal Involvement: A Case Report"

_brainsci, 2023, doi:10.3390/brainsci13020283_

Round 1
Reviewer 1 Report
Interesting and well documented case report of anti-MOG-antibody-associated encephalitis in young woman with rare unilateral (left-sided) cortico-subcortical involvement and bilateral meningeal affection. MOG was positive in CSF and serum. After IV dexamethasone, patient recovered, with normalization of MRT lesions and negative MOG. This unique case report adds to our knowledge about anti-MOG-AB-associated encephalitis. The essential literature is considered. Anothe rone could be added: Salama, S et al. 2019; Mult Scler 25,1427-33..
Reviewer 2 Report
The article titled "Anti-MOG antibody-associated unilateral cortical encephalitis with bilateral meningeal involvement: a case report" reported a rare case of anti-MOG antibody encephalitis (unilateral cortical encephalitis with bilateral meningeal involvement) in an adult patient. However, there are some problems to be solved.
Major points:
1) In figure 1, EEGs shows scatterd sharp wave, but the focus of the sharp wave was not understood only monopolar montage. You should show the bipolar montage about the same EEGs.
2) P.5,L.30, Discussion: Although the authors argue that the clinical symptoms of the patient included headache, disturbance of consciousness, aphasia, and fever, without optic nerve edema or spinal cord involvement, you have not show othe results of optic nerve function (ex. visual evoked potential (VEP)). The pattern-VEP is useful for early detection for optic nerve disorders and follow-up.
3) Although post-contrast MRI of the patient at admission (2 weeks after onset) showed revealed linear enhancement in frontoparietal, temporal and occipital sulci bilaterally, I couldn't find the reason for diagnosing the unilateral cortical encephalitis. Please explain the process of identifing the disease.
Reviewer 3 Report
Legend for figure 2 should be rewritten, as it contains duplicated words. In line to this, linear enhancement in frontoparietal, temporal and occipital sulci bilaterally should also be marked with arrows or circlec since the Authors compare the results to these obtained after one year.
The Authors writtent that a high dose of steroids is the first and recommended choice for MOG antibody-associated encephalitis. Therefore, my quiestion is why in this case a low dose was administered? An explanation should be provided. This choice suggests that the Authors twere unsure of the diagnosis
The Discussion section is quite superficial. The Authors should provide deeper discussion based on the literature available, and not simple compare the result.
Also, please insert some possible limitation of the study
Please provide also some information on medical history of the patient. It might be important since recently MOG antibodies encephalitis was found with related with COVID-19
Round 2
Reviewer 2 Report
Thank you for your information.
I understood all of them.
Reviewer 3 Report
The Authors have now provided sufficient responses and improved the paper. Therefore, it is now ready to be published.